# The role of oxygen-permeable ionomer for polymer electrolyte fuel cells

Ryosuke Jinnouchi[1 ✉], Kenji Kudo[1], Kensaku Kodama[1], Naoki Kitano[1], Takahisa Suzuki [1], Saori Minami[1], Kazuma Shinozaki[1], Naoki Hasegawa[1] & Akihiro Shinohara [1]

In recent years, considerable research and development efforts are devoted to improving the performance of polymer electrolyte fuel cells. However, the power density and catalytic activities of these energy conversion devices are still far from being satisfactory for large-scale operation. Here we report performance enhancement via incorporation, in the cathode catalyst layers, of a ring-structured backbone matrix into ionomers. Electrochemical characterizations of single cells and microelectrodes reveal that high power density is obtained using an ionomer with high oxygen solubility. The high solubility allows oxygen to permeate the ionomer/catalyst interface and react with protons and electrons on the catalyst surfaces. Furthermore, characterizations of single cells and single-crystal surfaces reveal that the oxygen reduction reaction activity is enhanced owing to the mitigation of catalyst poisoning by sulfonate anion groups. Molecular dynamics simulations indicate that both the high permeation and poisoning mitigation are due to the suppression of densely layered folding of polymer backbones near the catalyst surfaces by the incorporated ring-structured matrix. These experimental and theoretical observations demonstrate that ionomer's tailored molecular design promotes local oxygen transport and catalytic reactions.

[1] Toyota Central R&D Labs., Inc, Nagakute, Aichi, Japan. ✉email: jryosuke@mosk.tytlabs.co.jp

Polymer electrolyte fuel cells (PEFCs) are promising power sources for automobiles. Owing to intensive research and development of fuel cell systems and materials[1–6], several automakers have commercialized fuel cell vehicles (FCVs) with the required power density and durability[7,8]. However, they are still more expensive than vehicles with internal combustion engines. To reduce the system costs, it is essential to reduce the use of expensive materials and components, such as platinum, supports, membranes, ionomers, gas diffusion layers (GDLs), and bipolar plates[9]. An effective method of cost reduction is the use of PEFCs with high power density. Hundreds of cells are currently stacked in an FCV to ensure the availability of the maximum power required sporadic aggressive acceleration. Obtaining higher power density in individual cells allows automakers to reduce the stack cost by decreasing the number of cells or by reducing the geometric electrode area. In addition, the amount of precious Pt catalyst per cell needs to be reduced, and the cells must exhibit the required durability. Motivated by these demands, the US DRIVE Fuel Cell Tech Team has set a 2025 power density target of $1.8\ \mathrm{W\ cm^{-2}}$ [10], and the New Energy and Industrial Technology Development Organization in Japan recently set a 2030 power density target of $2.5\ \mathrm{W\ cm^{-2}}$ with $0.12$–$0.25\ \mathrm{mg\ cm^{-2}}$ Pt loading[11]. In addition, Million Mile Fuel Cell Truck (M2FCT) consortium has set a 2025 efficiency and durability target of $2.5\ \mathrm{kW\ g_{PGM}^{-1}}$ power ($1.07\ \mathrm{A\ cm^{-2}}$ current density) at 0.7 V after 25,000 hour-equivalent accelerated durability test for heavy-duty vehicle applications[12,13].

The cathode catalyst layers (CLs) of PEFCs are known to strongly influence fuel cell performance. Figure 1 shows a schematic of a single cell and CL. The CL is a porous medium composed of Pt or Pt alloy nanoparticles deposited on carbon supports and perfluorinated ionomers. Oxygen molecules diffuse through the pores in the CL and dissolve in the ionomers. The dissolved oxygen molecules permeate the ionomers and react with protons and electrons on the Pt surface to produce water. To meet the aforementioned performance targets, the CLs need to exhibit three key properties: high mass activity for the oxygen reduction reaction (ORR), high oxygen diffusivity, and high durability[14]. The Tafel slope of the ORR indicates that the voltage decreases by 60 mV owing to the decelerated kinetics of the ORR when the Pt loading is decreased to 1/10[15,16]. Therefore, high mass activity is necessary to decrease the Pt loading. However, electrochemical characterization of single cells[17–19] has revealed that high mass activity alone is not sufficient. For example, the characterizations by Greszler et al.[19] demonstrated that decreased Pt loading is accompanied by an anomalous voltage drop owing to oxygen transport resistance in the cathode CL. The measured resistance was found to be mathematically equivalent to that of the film-like resistive layer coated on the Pt surface. In this study, the resistive layer was attributed to the ionomers, which appear as films covering the Pt surface (Fig. 1). However, the origin of the high resistance of the ionomers remains controversial. The measured high resistance cannot be explained without assuming much thicker ionomer films and/or larger carbon agglomerates than those observed in the micrographs[19,20]. To elucidate the mechanistic origin of the anomalous voltage drop, intensive experiments and mathematical simulations have thus been undertaken. Kudo et al.[21,22] measured the oxygen transport resistance of Nafion thin films (see its molecular structure in Fig. 2a) using microelectrodes and observed a large interfacial resistance. On the basis of experimental results, Suzuki et al.[20] incorporated the interfacial resistance into a mathematical model and reproduced the experimental current–voltage curves reasonably well. Later, Liu et al.[23] showed negligible resistance at the gas/ionomer interface by experiments on Nafion thin films. In order to clarify the origin of the interfacial resistance, Jinnouchi et al.[24] and Kurihara et al.[25] performed molecular dynamics (MD) simulations and showed that the interfacial resistance originates from the dense ionomer layer created at the ionomer/Pt interface. By contrast, a more significant bulk ionomer resistance was recently proposed on the basis of hydrogen pump experiments[26] and a mesoscopic CL model[27] assuming highly reduced transport in thin films compared to that in bulk ionomers because of confinement effects[28] or substrate interactions[29].

Although the mechanism is not entirely clear, it is commonly accepted that oxygen transport resistance in the ionomer causes an unfavorable voltage drop. The simplest method of suppressing the voltage drop is to increase the specific Pt surface area, for example, by decreasing the Pt particle size, and thus reduce the local oxygen fluxes. However, this change has a detrimental effect on the durability because smaller particles degrade more rapidly by Ostwald ripening and particle coalescence[30–32]. An alternative method of solving this trade-off problem is to eliminate the ionomer/Pt contacts. Ionomer-free nanostructured thin films fabricated by 3M exhibit such desired environments. The ionomer/Pt interfaces can also be removed by using mesoporous carbon supports, where ionomers cannot cover the Pt nanoparticle catalysts located inside the mesopores[33–37]. However, in both cases, the proton conductivity decreases significantly in dry

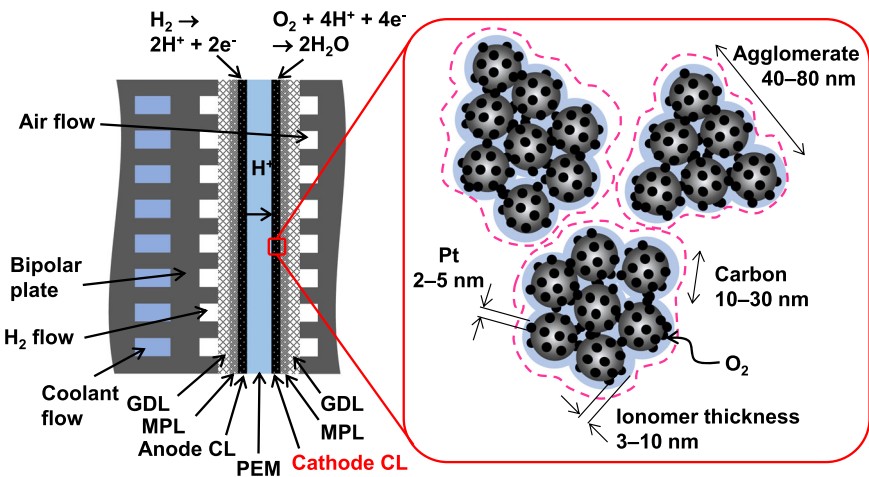

**Fig. 1 Schematic of PEFC and cathode CL.** PEM, MPL, and GDL denote the proton exchange membrane, microporous layer, and gas diffusion layer, respectively.

**Fig. 2 Molecular structures of ionomers. a–c** Molecular structures of Nafion ionomer (**a**), HOPI with PFMMD matrix reported in ref. [41] (**b**), and HOPI synthesized in this work (**c**). Density ($\rho$), number-average molecular weight ($M_n$), EW (ionomer mass per sulfonate group), and proton conductivity ($\sigma_H$) are also shown. The density was measured under dry conditions, and the proton conductivity shown here was obtained at 97% RH and room temperature.

operation because of the lack of proton conduction pathways in the absence of liquid water.

Several research groups have reported significant improvements in fuel cell performance without any remarkable adverse effects by using highly oxygen-permeable ionomers (HOPIs)[38–41]. For example, a recent study by Katzenberg et al.[41] showed that a HOPI incorporating an oxygen-permeable glassy amorphous matrix based on a perfluoro-(2-methylene-4-methyl-1,3-dioxo-lane) (PFMMD) backbone shown in Fig. 2b reduced the oxygen transport resistance in the CL and enhanced the power density of fuel cells. These results motivate the molecular design of ionomers to promote the local oxygen and proton transport required by the device architecture. For advanced design of cathode CL, however, further understanding of the role of the HOPI is necessary. The permeability is determined by both the bulk and interfacial permeabilities, which are strongly affected by multiple properties such as the oxygen solubility, diffusivity, interfacial permeation rate constants, and ionomer distribution in the CL[20–22,24,42]. Changes in the interfacial structure caused by the new ionomers can also influence the ORR activity[43–46]. Clarification of these relevant effects is significant for optimal arrangement of catalysts, ionomers, and support materials in the CL as well as molecular design of ionomers.

In this work, by combining analyses of single cells, microelectrodes, and single-crystal surfaces with MD simulations, we show that an HOPI incorporating a ring-structured monomer, perfluoro-(2,2-dimethyl-1,3-dioxole) (PDD)[47], significantly enhances both the interfacial oxygen permeation and ORR activity. The high permeation is shown to originate from the high oxygen solubility, and the high ORR activity is attributed to the mitigation of catalyst poisoning by sulfonate anion adsorption. All these improvements originate from the PDD matrix, which prevents layered folding of the ionomer backbones on the Pt surface.

## Results and discussion

Figure 2c shows the molecular structure of our HOPI. Its $^{19}F$ Nuclear Magnetic Resonance (NMR) spectrum is compared to those of the Nafion ionomer and HOPI in Supplementary Fig. 2a. The peak assignments of the NMR spectra were carried out using the values reported in past studies[48–50]. Our HOPI contains a symmetric PDD matrix, which is expected to introduce highly oxygen-permeable amorphous domains similar to those in the PFMMD matrix[41]. The obtained $^{19}F$-NMR spectra indicate the presence of the perfluorinated sulfonic-acid (PFSA)[48] matrix and PDD matrix[49,50] in the synthesized HOPI ionomer. A previous study[41] showed that an ionomer with PFMMD backbones had a low density of $1.86–1.89\,g\,cm^{-3}$ under dry conditions compared to that of Nafion ($2.04\,g\,cm^{-3}$). The density of our HOPI is also lower than that of Nafion ($1.93\,g\,cm^{-3}$). The number-average molecular weights ($M_n$) of the Nafion ionomer and HOPI were $2.8 \times 10^4$ and $3.9 \times 10^4$, respectively, and the equivalent weights

(EWs) were 952 and $735\,g\,mol^{-1}$, respectively. The measured $M_n$ of the Nafion ionomer is smaller than the reported value[51] because of the difference in the standards used in the calibration (see details of the calibration in SI). The relative difference between Nafion and HOPI in our study indicates that the molecular weight of the HOPI does not significantly differ from that of Nafion. Although the EW of the HOPI is lower, the HOPI does not differ greatly in proton conductivity and water uptake from Nafion, as shown in Supplementary Fig. 3. This is due to the presence of the amorphous PDD domains, which do not contribute to proton conduction[41]. As shown in Supplementary Fig. 2c, the wide-angle X-ray scattering (WAXS) indicates that the HOPI does not exhibit the crystalline matrix peak at $0.8–1.4\,Å^{-1}$; the PDD domains presumably suppress the backbone crystallization as in the case of the PFMMD domains[41].

Figure 3a shows the current–voltage curves of membrane electrode assemblies (MEAs) with the Nafion ionomer and HOPI. The HOPI outperforms Nafion over the entire current density range at 30% and 60% relative humidity (RH) and high current density range at 90% RH. The improvement under high-current-density conditions is attributed to the reduction in the local oxygen transport resistance $R_{other}$ by the HOPI. As shown in Fig. 3b, the $R_{other}$ value of the MEA employing HOPI is smaller in comparison to Nafion, particularly at 30% RH. The $R_{other}$ value of Nafion increases significantly when the RH decreases from 100% to 30%, indicating that the dry Nafion forms highly resistive films on the Pt surfaces. By contrast, the HOPI retains a low $R_{other}$ under low-RH conditions. Unlike $R_{other}$, as shown in Fig. 3c, the HOPI exhibits ohmic resistance of the CL higher than Nafion. Hence, the observed improvement in the current–voltage performance is attributed to the lower local oxygen transport resistance of the HOPI. The reduction in $R_{other}$ was also observed for the HOPI with the PFMMD domains[41]. Detailed comparison indicates that our HOPI exhibits more significant reduction (40–50% reduction) to $R_{other}$ than the PFMMD-based HOPI (17% reduction).

Our previous analyses using microelectrode techniques[21,22,24] indicated that the local oxygen transport resistance in the CL with the Nafion ionomer is dominated by the interfacial permeation resistance of the ionomer thin film. The current study also suggests the same conclusion. Figure 4a shows the inverse of the measured limiting current density as a function of the ionomer thickness. The inversed current density approximately obeys the linear Eq. (2), and the slope and intercept provide the bulk and interfacial gas diffusion resistances, respectively (see details in the "Methods"). As summarized in Fig. 4b, the interfacial resistance of the Nafion ionomer (black solid line) is one order of magnitude higher than the bulk resistance (black dashed line). Hence, the small $R_{other}$ of the MEA with the HOPI must be attributed to the enhanced interfacial oxygen permeation. Indeed, the interfacial resistance of the HOPI (red solid line) is only one-fifth to one-half the resistance of Nafion (black solid line), as shown in

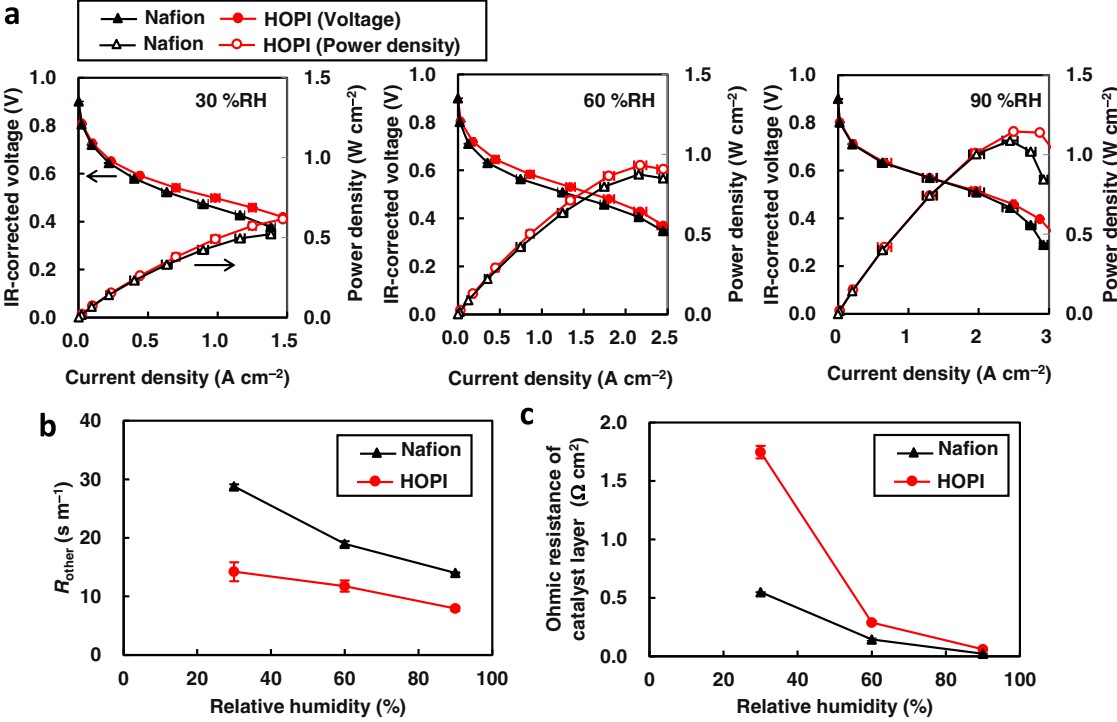

**Fig. 3 Electrochemical characterizations of MEAs. a** Current–voltage (with closed dots) and current–power density (with open dots) curves of MEAs with Nafion ionomer (black) and HOPI (red) at 353 K and 30, 60, and 90% RH. **b** Local oxygen transport resistance $R_{other}$ as a function of RH. **c** Ohmic resistance of CLs in MEAs. All measurements were conducted twice, and the differences between two measurements were shown as error bars in all figures.

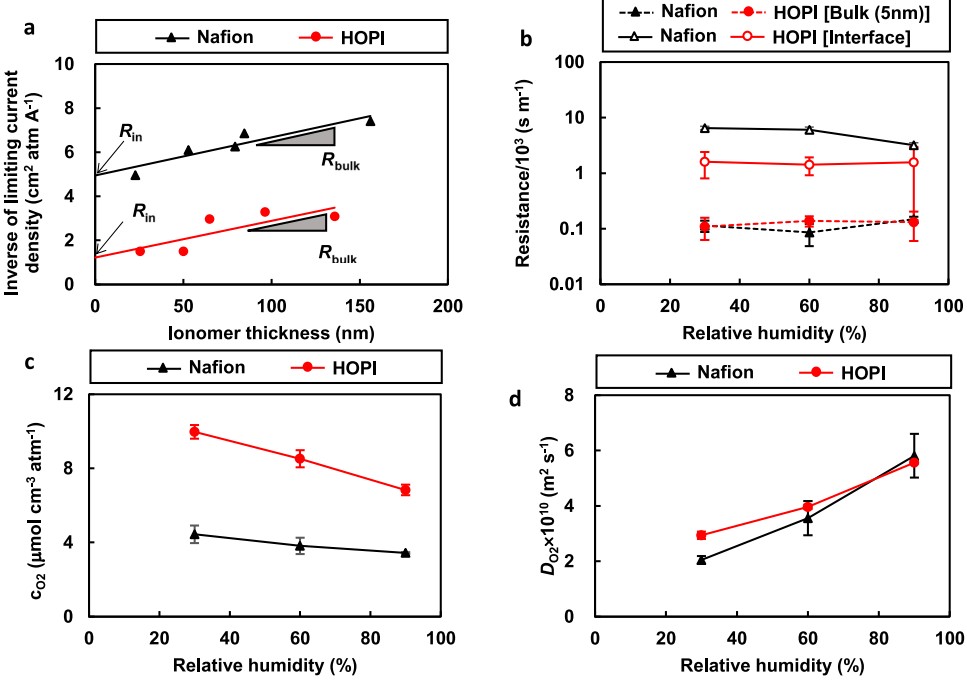

**Fig. 4 Electrochemical characterizations of microelectrodes. a** Inverse of the limiting current density as a function of the ionomer thickness at 333 K and 30% RH. The bulk ($R_{bulk}$) and interfacial ($R_{in}$) resistances are determined by the slope and intercept of the linear line, respectively. **b** Interfacial (open symbols with solid lines) and bulk (solid symbols with dashed lines) oxygen permeation resistance of ionomer thin films measured by the microelectrode technique. **c, d** Oxygen solubility and diffusion coefficient of ionomer thick films measured by the microelectrode technique. In **b**, the bulk resistance was calculated assuming an ionomer thickness of 5 nm. The error bars shown in (**b**) were calculated from fitting errors assuming the normal error distributions, and the error bars shown in (**c**) and (**d**) show the differences between two measurements.

Fig. 4b. The analytical equation for oxygen diffusion [Eq. (2) in the "Methods"] indicates that the smaller interfacial resistance results from the higher oxygen solubility (or Henry constant $K_{O2}$) in the HOPI and/or the higher interfacial permeation rate constants ($k_{ion/Pt}$ and/or $k_{ion/gas}$). The contributions of the solubility and permeation of the ionomer thick film can be separated using the microelectrode technique (see the "Methods"). The measured oxygen solubility and diffusion coefficient are summarized in Fig. 4c and d, respectively. The solubility in the HOPI is two to three times higher than that in Nafion, whereas the diffusion coefficient does not differ notably. This result indicates that the approximately threefold increase in the oxygen permeation is attributable to the difference in oxygen solubility, and the remaining increase is attributable to the differences in permeation rate constants. The rate of the permeability increase realized by our HOPI is close to the result reported in the past study on the HOPI with the PFMMD matrix[41] and is higher than the result reported on the HOPI with another type of ring-structured backbone[40]. Permeabilities reported in the literatures[52,53] indicate that TeflonAF composed only of the PDD matrix exhibits a remarkably high oxygen permeability while Aquivion with short sidechains (see its structure in Supplementary Fig. 2c) does not exhibit notable change. Therefore, the high permeability of our HOPI is attributed to the ring-structured PDD domains.

The high oxygen permeability of HOPI is investigated via MD simulations. The simulated oxygen permeation resistance of the HOPI is less than one-third that of Nafion (Fig. 5a). As shown in Fig. 5b and c, the oxygen solubility of the HOPI is higher than that of Nafion particularly at low-RH conditions while the oxygen diffusion coefficient of the HOPI is comparable to or lower than that of Nafion. Although the simulations exhibit quantitative differences in the permeation resistance and diffusion coefficient (Note: discrepancies in experimental and simulated results arise from approximated inter-atomic potential models and limited simulation time), both the simulations and experiments indicate that the enhanced solubility greatly contributes to the high interfacial permeability. The high solubility in the HOPI results from its low density. The MD simulations indicate that the bulk HOPI has a lower density ($1.90 \pm 0.01\,\mathrm{g\,cm^{-3}}$) than bulk Nafion ($2.01 \pm 0.01\,\mathrm{g\,cm^{-3}}$), as in the experimental result (see Fig. 2). A more pronounced trend is observed in the simulated density profiles of the ionomer thin films (Fig. 5d), where the average density of the HOPI thin film was calculated to be roughly 85% of that of the Nafion ionomer thin film. Near the Pt surface in particular ($Z = 0.3$–$0.67\,\mathrm{nm}$), the density of the HOPI is low. The MD snapshots and density profiles indicate that the polytetrafluoroethylene (PTFE) backbones of the Nafion ionomer are folded to form a layered structure parallel to the Pt surface. By contrast, the HOPI backbones form a more disordered structure. Owing to the layered folding, the Nafion ionomer thin film becomes 10% thinner than the HOPI thin film. In addition, the local density of the Nafion adlayer near the Pt surface ($0.3$–$0.67\,\mathrm{nm}$ from the Pt surface) is 20% higher than the density of the HOPI in the same region. As demonstrated in our previous study on the Nafion ionomer[24], the dense ionomer layer significantly reduces the oxygen solubility and increases the permeation barrier at the interface between Pt and ionomer. Accordingly, the PDD matrix prevents backbone folding and enhances the interfacial oxygen permeability near the Pt surface. Another notable conclusion provided by our MD simulations is that the permeation through the ionomer/Pt interface remains the rate limiting process in the HOPI. This is demonstrated by the oxygen distributions illustrated in Fig. 5e, where oxygen depletion layers appear at the ionomer/Pt interfaces in both Nafion and HOPI. The simulated result is consistent with the experimental

result shown in Fig. 4b where the interfacial resistance of HOPI is demonstrated to be larger than the bulk resistance.

The HOPI enhances not only the oxygen permeability but also the ORR activity. As shown in Fig. 6a, the mass ORR activity in the MEA is enhanced by the HOPI ionomer. The same trend was observed for the specific ORR activity on the Pt(111) single-crystal surface, as shown in Fig. 6a and b. This enhancement is caused by the mitigation of catalyst poisoning by sulfonate anion adsorption on the Pt surface. The mitigation is clearly visible in the cyclic voltammograms (CVs) shown in Fig. 6c. In the same figure, the cyclic voltammogram of the Pt(111) surface without ionomer is also shown as the reference. The oxidation charge at 0.4–0.5 V, which is not observed in the reference voltammogram, is attributed to the adsorption of sulfonate anions[45]. The integrated charge indicates that the surface coverage of sulfonate anions in the HOPI is 0.05 monolayers smaller than that in Nafion, as shown in Fig. 6d. Although the coverage difference between Nafion and the HOPI seems small, the mitigated anion adsorptivity greatly improves the ORR activity because the adsorbed sulfonate group brings another poisoning moiety, the ether in the sidechain, close to the Pt surface, as discussed in our previous study[46]. Comparison with Aquivion indicates that both the PDD matrix and short sidechain greatly contribute to the ORR activity and sulfonate surface coverage. Figure 5f illustrates that our MD simulation reproduces the decreased sulfonate anion coverage of the HOPI. The simulated density profile and experimental voltammograms indicate that the decreased anion coverage is due to limited mobility of sulfonate anion groups caused by the suppression of the layered backbone folding and the shorter sidechain structure.

In summary, analyses combining the electrochemical measurements of single cells, microelectrodes, and single-crystal surfaces with MD simulations elucidated the role of the HOPI in PEFCs. The HOPI was found to simultaneously enhance the oxygen transport and mass ORR activity in the cathode CL. Detailed microelectrode analyses and MD simulations showed that the high permeation in the HOPI is attributed mainly to the high oxygen solubility, which enhances the oxygen permeation through the ionomer/Pt interface. The MD simulations also indicated that the interfacial oxygen permeation remains a bottleneck of the oxygen transport in the HOPI. Electrochemical measurements using single-crystal electrodes indicated that high ORR activity is realized by the mitigation of catalyst poisoning by sulfonate anions. MD simulations give evidence that both effects arise from the ring-structured backbone matrix, which prevents the layered backbone folding of the ionomer near the catalyst surface. The experimental and theoretical observations presented herein indicate that a tailored molecular design of the ionomer makes it possible to simultaneously realize high interfacial oxygen transport and high ORR activity.

## Methods

**Ionomer synthesis**. The ionomer was synthesized in two steps, polymerization and hydrolysis, as shown schematically in Supplementary Fig. 1. PDD (97%, P&M) and perfluoro(3-oxapent-4-ene) sulfonyl fluoride (PSVE) (99%, SynQuest) were distilled prior to use. Polymerization was performed under inert atmosphere by stirring a mixture of PDD (4.0 g, 16.4 mmol), PSVE (13.7 g, 48.9 mmol), and a solution (0.08 cm$^3$, 0.08 mol dm$^{-3}$) of a polymerization initiator, $[\mathrm{CF_3(CF_2)_2C(=O)O-}]_2$, prepared from $\mathrm{CF_3(CF_2)_2C(=O)Cl}$ (Tokyo Chemical Industry Co., Ltd) in a Vertrel$^{\mathrm{TM}}$ XF solvent (DuPont-Mitsui Fluorochemicals Co., Ltd) for 3 days at 288 K. The remaining monomers were removed by heating the product liquid at 373 K after polymerization, and the copolymerized polymer with the side-chain end of the SO$_2$F group (2.5 g) was obtained. The copolymerized polymer (1.03 g) was mixed with a NaOH aqueous solution (1 mol dm$^{-3}$) in a Teflon crucible and hydrolyzed by heating the solution at 403 K for 12 h. After the solvent was removed, the residual solid polymer was immersed in a HCl aqueous solution (1 mol dm$^{-3}$) and heated at 353 K. After the polymer was washed with ultrapure water several times and dried at 353 K, an ionomer with the side-

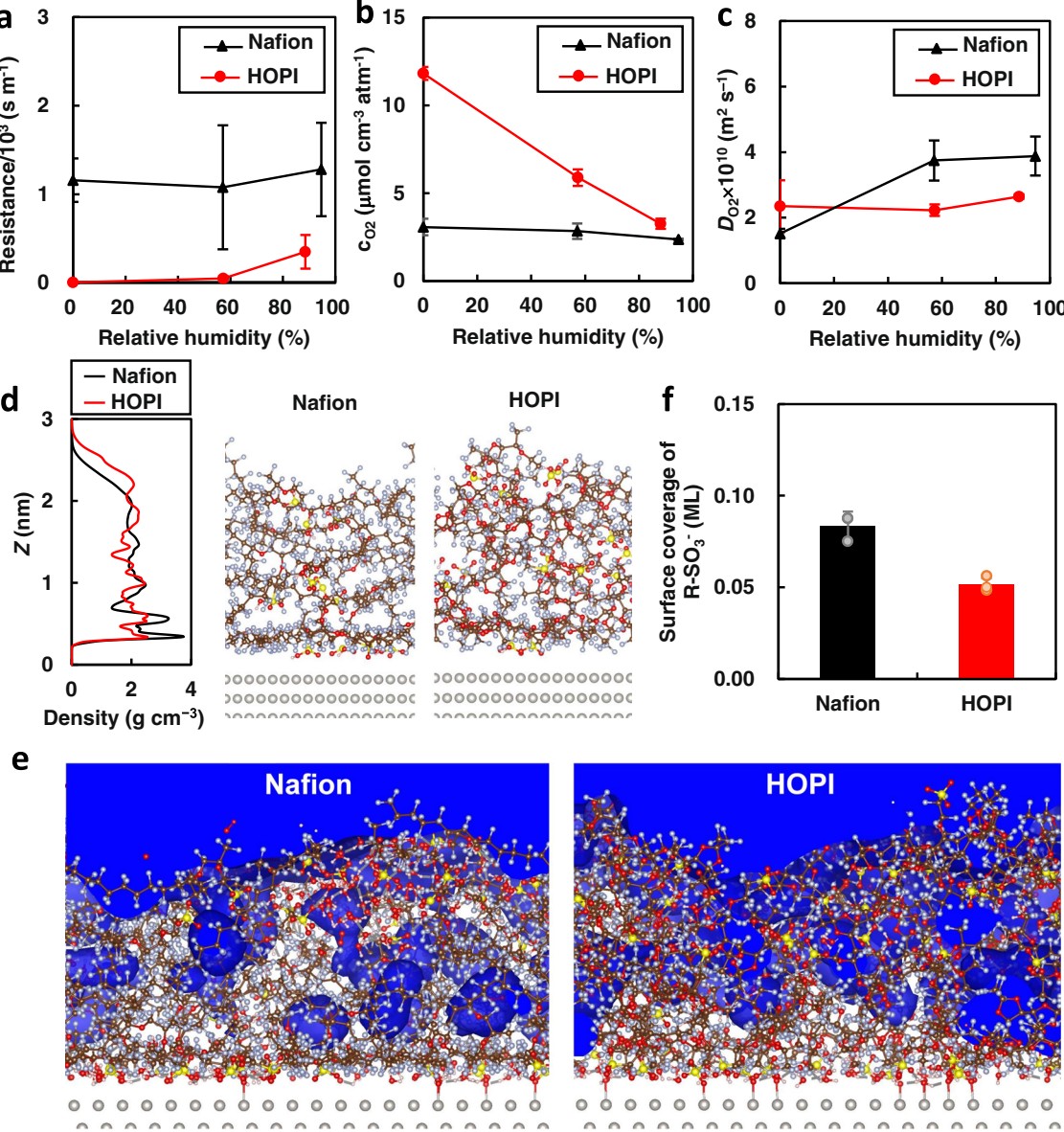

**Fig. 5 Bulk and interfacial properties predicted by MD simulations. a–c** Simulated oxygen permeation resistance (**a**), oxygen solubility (**b**), and oxygen diffusion coefficient (**c**) in Nafion (black) and HOPI (red). **d** Density distributions of the ionomer thin films and snapshots of the interfacial structures at RH = 0% ($\lambda = 0$). **e** 3-dimensional oxygen distributions at RH ≈ 60%. The region with $O_2$ concentration higher than 7 μmol $cm^{-3}$ $atm^{-1}$ is shown in blue. **f** Surface coverage of sulfonate anions in the ionomers determined from the MD simulations (RH ≈ 60%, $\lambda = 6$). The surface coverage was determined from the average number of sulfur atoms present at 0–4.5 Å from the Pt surface. In all figures, error bars indicate the root means square differences of two to three MD runs started from different initial structures (see details in the "Methods"). White, red, brown, light blue, yellow, and gray spheres in the molecular structures made by the VESTA software[60] indicate H, O, C, F, S, and Pt atoms, respectively.

chain end of the SO₃H group (0.86 g) was obtained. After synthesis, the molecular weight, ¹⁹F-NMR, WAXS, EW, density, water uptake, and proton conductivity were measured. The measurement methods are summarized in Supplementary Information Section 2.

**Membrane electrode assembly**. A Pt/Vulcan catalyst (TEC10V30E, Tanaka Kikinzoku Kogyo, Tokyo, Japan), an ionomer solution (Nafion D2020, Chemours, or HOPI), and a solvent were mixed and agitated by ultrasonic vibration so that the final solvent composition became water/ethanol/propanol 0.50/0.44/0.06 weight ratio for Nafion and water/ethanol/propanol/propylene glycol 0.50/0.02/0.38/0.10 weight ratio for HOPI. In the HOPI solution, propylene glycol was mixed to prevent formations of cracks in the catalyst layer, and propanol weight ratio was increased to dissolve the ionomer homogeneously. The ionomer/carbon weight ratio was 0.75. This ink was applied to a PTFE sheet by a doctor blade process and dried to form a catalyst layer sheet with a Pt loading of 0.12 mg_P cm⁻². This catalyst sheet (1 cm²) was transferred to a Nafion membrane (NR-211) by a decal method using hot pressing (60 kgf cm⁻², 413 K, 5 min). An anode CL was formed

similarly using Pt/Vulcan at 0.2 mg_Pt cm⁻² and an ionomer/carbon weight ratio of 0.75 to form an MEA. A single cell was assembled using the MEA and GDL substrates (TGP-H-030, Toray) with microporous layers and gold-plated copper current collectors with a straight channel flow field (0.4-mm-wide channels and lanes).

Electrochemical measurements of the single cells were performed using protocols similar to those recommended by Fuel Cell Commercialization Conference of Japan (FCCJ)[54] and EU harmonized testing protocols[55]. Detailed conditions were modified to adapt the protocols to our single cell and targets of this study. Details are described in Supplementary Information Section 3. All measurements were carried out using a charge/discharge unit (Hokuto Denko, Japan) with a fuel cell testing system (Chino Co., Tokyo, Japan). The reactant gas (hydrogen or oxygen) and nitrogen were mixed and humidified by bubbling through water at controlled temperatures. The current–voltage characterizations of the MEA are described in detail in Supplementary Information Section 3. The local oxygen transport resistance in the MEA was measured using the method proposed by Ono et al.[18] In this method, the total oxygen transport resistance $R_{total}$ at the limiting current defined by Baker et al.[56] is separated into the molecular diffusion

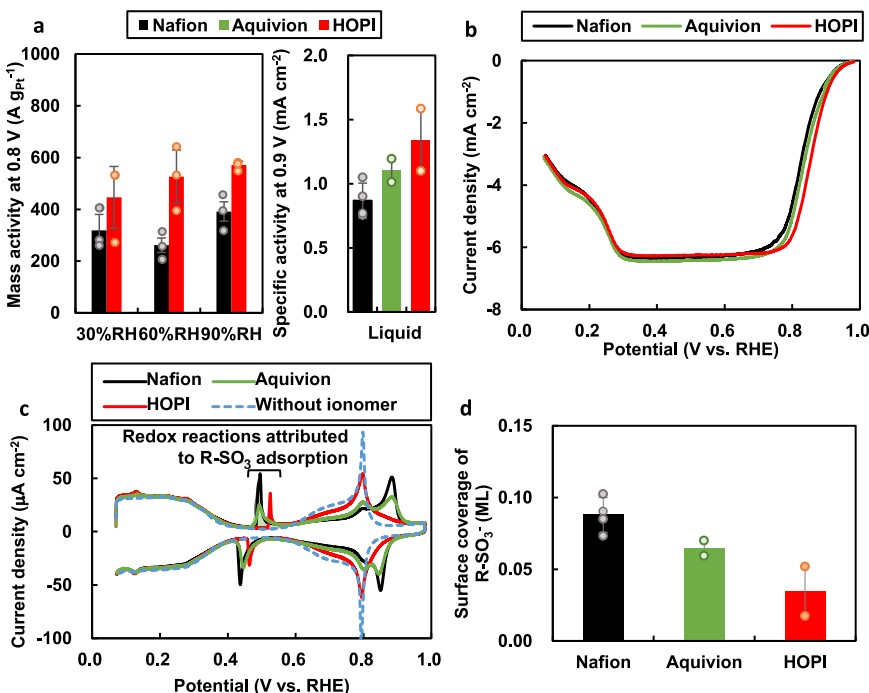

**Fig. 6 Electrochemical characterizations of single-crystal electrodes. a** Mass ORR activity in the MEAs with the Nafion ionomer and HOPI (left), and specific activity of the Pt(111) surfaces covered by the Nafion ionomer, Aquivion ionomer, and HOPI (right). **b, c** Linear-sweep voltammograms and CVs of the Pt(111) surfaces covered by the Nafion ionomer, Aquivion ionomer, and HOPI under oxygen-saturated and inert conditions, respectively. **d** Surface coverage of sulfonate anions in the ionomers determined from the adsorption charges in the CVs. The adsorption charges were determined by subtracting the background charges calculated from the CV in a 0.1 mol dm$^{-3}$ HClO$_4$ solution without an ionomer (blue dashed line in **c**). The plots in bar graphs are the measured data, and error bars show their root mean square differences. Bars show their averages based on two to four separate measurements.

resistance $R_{mol}$ and remaining resistance $R_{other}$:

$$R_{total} = \frac{4Fc_{O_2}}{j_L} = R_{mol}\frac{p}{p_0} + R_{other}, \qquad (1)$$

where $F$ is the Faraday constant (C mol$^{-1}$), $c_{O_2}$ is the concentration of oxygen molecule (mol m$^{-3}$), $j_L$ is the limiting current density (A m$^{-2}$), $p$ is the total pressure (Pa), and $p_0$ is the reference total pressure (0.1 MPa). The first resistance in Eq. (1) is attributed mainly to intermolecular diffusion in the GDL, which possesses relatively large pores, where diffusion coefficients are inversely proportional to the total pressure. The second resistance is attributed to the diffusion in the CL having small pores and thin ionomer, where the gas transport is dominated by pressure-independent Knudsen diffusion and diffusion in the ionomer and liquid water[21,22,24,26,27]. To determine $R_{other}$, the limiting current density $j_L$ of the MEA was measured by varying the total pressure $p$. $R_{mol}$ and $R_{other}$ were determined as the slope and intercept, respectively, of a plot of $R_{total}$ versus $p$. The measurement is explained further in Supplementary Information Section 3. In addition to the local oxygen transport resistance, the ohmic resistance of the proton exchange membrane (PEM) and CLs was measured by the AC impedance method[57]. The experimental conditions are also described in Supplementary Information Section 3.

**Microelectrodes.** The interfacial oxygen permeation resistance of the ionomer was measured using the microelectrode technique developed in ref. [21]. Assuming the presence of constant bulk and interfacial oxygen permeation resistances through the ionomer, the limiting current density $j_L$ of the ORR should obey the following equation:

$$\frac{4Fc_{O_2}}{j_L} = \frac{1}{RTD_{O_2}K_{O_2}}\delta + \frac{1}{RTK_{O_2}}\left(\frac{1}{k_{ion/Pt}} + \frac{1}{k_{ion/gas}}\right), \qquad (2)$$

where $R$ is the gas constant (J K$^{-1}$ mol$^{-1}$), $T$ is the temperature (K), $D_{O_2}$ is the diffusion coefficient of oxygen molecule in the bulk ionomer (m$^2$ s$^{-1}$), $K_{O_2}$ is the Henry constant (mol m$^{-3}$ Pa$^{-1}$), $\delta$ is the ionomer thickness (m), and $k_{ion/Pt}$ and $k_{ion/gas}$ are the apparent rate constants of oxygen permeation through the ionomer/Pt and ionomer/gas interfaces (m s$^{-1}$), respectively. The bulk resistance coefficient $R^{-1}T^{-1}D_{O_2}^{-1}K_{O_2}^{-1}$ and the interfacial resistance $R^{-1}T^{-1}K_{O_2}^{-1}(k_{ion/Pt}^{-1} + k_{ion/gas}^{-1})$ can be determined from the slope and intercept, respectively, of a plot of $j_L^{-1}$ versus $\delta$. As discussed in our previous study[21], the ionomer must be thin enough to resemble the ionomer thin films in the CL and to enable accurate determination of the intercept. The sample preparation and measurement are described in Supplementary Information Section 4.

Although the thin film measurement provides the bulk and interfacial resistances, these parameters both depend on multiple properties of the ionomer: the solubility, diffusivity, and interfacial permeation rate constants. For further separation, we measured the limiting current transition of the ORR, which flows through the interface between the ionomer thick film and the Pt microelectrode after the potential step measurements under oxygen atmosphere. When the potential is stepped from 1.1 to 0.4 V (vs. RHE), for example, the transitional limiting current controlled by oxygen transport inside the thick ionomer film is observed. Assuming that the ionomer film is thick enough that boundary effects are negligible, the time-dependent limiting current density can be described by the following Cottrell-type equation[53,58]:

$$j_L(t) = \frac{4Fp_{O_2}K_{O_2}D_{O_2}}{R_0}\left[1 + \frac{R_0}{\sqrt{\pi D_{O_2}t}} + 0.2732\exp\left(-\frac{0.3911R_0}{\sqrt{\pi D_{O_2}t}}\right)\right], \qquad (3)$$

where $t$ is the time from the potential step, and $R_0$ is the radius of the microelectrode. $K_{O_2}$ and $D_{O_2}$ can be determined by fitting Eq. (3) to the measured transitional limiting current density. The sample preparation and measurement are also described in further detail in Supplementary Information Section 4.

**Single-crystal model electrodes.** A (111)-oriented Pt single-crystal disk (99.99%, 0.196 cm$^2$, MaTecK) was annealed by electromagnetic inductive heating for more than 10 min at 1400–1650 K in a mixed H$_2$ and Ar flow (3% H$_2$, Taiyo Nippon Sanso; H$_2$: 99.99999%, Ar: 99.999%). The annealed specimen was slowly cooled to room temperature in a flow of the same mixed gas, and the Pt(111) surface was covered by a droplet of ultrapure water (Milli-Q, 18.2 MΩ). The Pt(111) surface was then coated with a thin film of an ionomer by dropping 20 μL of a 0.005 wt% ionomer solution (20 wt% dimethylformamide aqueous solution), and the solvent was evaporated from the surface in an Ar stream; finally, the film was heated at 420 K in the Ar stream to improve its physicomechanical stability. Further details are given elsewhere[43]. Equal amounts by weight of the ionomers were deposited on the Pt(111) surface to a thickness of ~35 nm. Cyclic and linear-sweep (positive scan) voltammetries were carried out in a 0.1 mol dm$^{-3}$ HClO$_4$ aqueous solution under inert and oxygen-saturated conditions, respectively, at a scan rate of 50 mV s$^{-1}$. The temperature was set to 303 K. The electrode was rotated at 1600 rpm in the linear-sweep voltammetry.

**Molecular dynamics simulations.** Nonequilibrium MD simulations[24] were employed to compute the permeation flux of oxygen through the ionomer thin films on the Pt(111) surface. In the simulations, a high-pressure (20 MPa) pure

oxygen gas was applied to the ionomer thin film. During the MD simulation, some oxygen molecules permeated the ionomer and reached the Pt surface. By translating these oxygen molecules back to the upper limit of the gaseous oxygen layer, steady-state oxygen permeation was obtained during the MD simulation. From the number of transferred oxygen molecules, the oxygen permeation flux was calculated and directly compared with the experimentally measured limiting current density. During the MD simulations, the temperature was maintained at 353 K by an *NVT* ensemble realized by the weak coupling method[59]. For each condition, three initial structures were prepared, and a 150 ns MD simulation was executed for each initial structure to calculate the oxygen permeation flux. The obtained fluxes were averaged and used to calculate the limiting current density. In addition to the ionomer/Pt interfaces, the bulk ionomers were simulated to compute the ionomer densities and diffusion coefficients of oxygen molecule in the ionomers. Oxygen molecules were located in the bulk ionomer models, and the diffusion coefficients were calculated from the slopes of the mean square displacements versus time. The ionomer density was calculated from the system volume. Similar to the ionomer/Pt interfaces, three initial structures were prepared, and a 10 ns MD simulation for each structure was executed. Obtained densities and diffusion coefficients were averaged. Ionomer thin films isolated in the high-pressure oxygen gas (20 MPa) were also simulated to calculate the solubility of oxygen in the thin films. Two initial structures were prepared for each condition, and a 30 ns MD simulation at 353 K for each structure was carried out. The solubility was calculated by counting the number of oxygen molecules in the region where the ionomer density is higher than 90% of the bulk density. Further details of the system size, model preparation, and simulations are described in ref.[24] and Supplementary Information Section 5.

**Reporting summary**. Further information on research design is available in the Nature Research Reporting Summary linked to this article.

## Data availability

All data are available in the main text and Supplementary Information. Additional datasets related to this study are available from the corresponding author upon reasonable request. All electrochemical, spectroscopic, and simulation source data generated in this study are provided in the Supplementary Information/Source Data file. Source data are provided with this paper.

## Code availability

The molecular dynamics simulation code is a software developed internally in Toyota Central R&D Labs., Inc. and is not publicly available.

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

## Acknowledgements

We would like to thank Dr. Masashi Harada for the WAXS measurements and Dr. Ryuichi Murase for his careful proofreading of the manuscript.

## Author contributions

A.S. synthesized a HOPI and measured its physical properties through discussion with N.H. N.K. and S.M. fabricated the MEAs and performed the electrochemical characterizations. Kenji Kudo developed the microelectrode techniques. Kenji Kudo and N.K. fabricated thick and thin ionomer films and performed the microelectrode measurements. Kensaku Kodama performed the electrochemical characterizations of single-crystal electrodes. R.J. conducted the MD simulations, analyzed the experimental and theoretical data through discussing with N.K., Kenji Kudo, Kensaku Kodama, K.S. and T.S. R.J. and A.S. wrote the manuscript. All authors discussed the results and checked the manuscript.

## Competing interests

The authors declare no competing interests.
