## [Peer Review File · Nature Communications]

REVIEWER COMMENTS

Reviewer #1 (Remarks to the Author):

In this manuscript the authors report a novel perfluorinated ionomer for polymer electrolyte membrane fuel cells featuring a ring-structure in the polymer backbone. The manuscript includes characterisation of the polymer's functional properties (molecular weight, equivalent weight, ...); current-voltage curves of operating cells; microelectrode experiments to determine oxygen diffusion properties; single crystal electrode studies to investigate surface interactions and molecular dynamic simulations of ionomers at the surface.

The manuscript makes the following major claims:

- That the ionomer has a higher oxygen permeability (predominantly solubility) than a traditional Nafion ionomer reducing local oxygen mass transport resistance with the main effect being at the boundaries of the ionomer thin films;
- That the ionomer exhibits less sulfonate ion poisoning of a platinum surface than a traditional Nafion ionomer increasing mass activity of the catalyst;
- That these effects are caused by the inclusion of the ring-structure in the polymer backbone which prevents layered backbone folding.

The ionomer appears novel and its increased permeability and resulting increase in cell performance when compared to Nafion, still a state-of-the-art ionomer, is impressive. As the authors highlight, other groups have developed other highly oxygen permeable ionomers (HOPI) e.g. Ref 36 [A. Rolfi] and Ref 37 [A. Katzenberg et. al.] in which the authors incorporate an alternative cyclic group into the backbone of an ionomer and make similar claims around disruption of polymer crystallinity and resulting improved oxygen permeability. Nevertheless, this work makes a substantial contribution to the development of fuel cell ionomers and contributes to the ongoing discussion around the origins of local mass transport resistance in fuel cell catalyst layers. Both of these areas are of great topical importance in the development of PEMFC. It is likely therefore, to be of great interest to both the ionomer and catalyst layer community and the wider PEMFC and electrocatalyst fields. In my opinion this work is likely, with the support other recent publications in the area, to influence the community to further develop HOPIs in order to improve the performance of PEMFC.

Generally, the manuscript supports its major claims in a convincing way. The MEA experiments reported show cells operating with a higher performance and lower local oxygen transport resistance. The microelectrode experiments performed support this, showing much lower interfacial oxygen permeation resistance for HOPI than Nafion. This is evident both in the calculated permeation resistances and the raw data. Single crystal LSVs experiments in oxygen and nitrogen show clear evidence that Pt(111) is more active when covered with HOPI than Nafion and that REDOX processes on the surface change.

The hypothesis that the incorporation of the ring structure alters the folding of the polymer backbone and thereby impacts the oxygen transport at the Pt-Ionomer interface is sensible; however I don't have the required background in MD to understand if the simulation presented here has will have high enough fidelity for the claim to be fully supported without further corroborating evidence. The evidence that changes to the backbone of the polymer are responsible for changes in absorption onto the surface is slightly tenuous. Compared to Nafion both the backbone and sidechain of HOPI have changed and either could conceivably alter the sidechain/sulfonate interaction with the surface.

Though the manuscript generally supports its conclusions well there are a few issues that the authors should address:

- There is no chemical or direct physical characterisation of the produced polymer provided in the supporting information. It's therefore not clear how the authors directly confirmed that a co-polymer is produced. Furthermore, there is no information on the degree or type of co-polymerisation or evidence of the crystalline structure of the polymer which would support the MD simulations.

- In the HOPI the sulfonic acid side chain is different from that in Nafion. Such short side chain ionomers / membranes are well known e.g. 3M and Aquivion ionomers and the properties are known to differ substantially from Nafion. It's therefore somewhat unclear to what extent improvements (especially relating to interaction with the Pt-111 surface) are caused by changes to the sidechain vs the backbone. Comparison experiments (or at least a literature comparison of properties such as oxygen solubility) between the performance of HOPI and an equivalent short side chain ionomer would be valuable.
- Generally, the discussion would benefit from links to relevant literature so that measured values, for instance permeability, are presented in the context of other work (e.g. other reports of measured properties on Nafion, Aquivion and other HOPI) to highlight the relative significance of the improvement reported.
- Many of the experimental properties of the ionomer reported, such as the mass transport resistance, solubility and diffusion coefficient, are reported without error bars or other indication of uncertainty. There is no indication that repeats were performed nor is reproducibility of the microelectrode experimental technique discussed.
- The electrochemical characterisation performed on MEA is carried out potentiodynamically with rather high sweep rates (20 mV s⁻¹). The vast majority of MEA testing is carried out under steady state/pseudo-steady state conditions as this allows time for the cell to equilibrate. Ideally MEA testing would be repeated under conditions more normal for cell testing.

A few additional minor comments and questions the authors may wish to consider:

- There is some key experimental information missing, though presumably available in the author's earlier works, for instance the geometry of the microelectrode and information on electrode conditioning and electrolyte used in measurements. It's recommended the authors review the experimental sections reported add some key information to the SI if space permits.
- The quality of text in Fig 4 is low in my copy, please check it's uploaded correctly.
- When measuring water uptake (λ) please provide more experimental details and information on data processing (any averaging of up and down sweeps etc.).
- For MEA testing please report all relevant conditions, e.g. those identified in the JRC's Fuel Cell Harmonised Testing Protocol.
- For measurements of local oxygen transport resistance: how is the partial pressure of the oxygen adjusted? Partial pressures should ideally be reported as pressures in MPa or kPa not %.
- In Sup Fig 5 Nyquist plots should be on isometric axes.
- In Sup Fig 7 more information about the quality of the fitting and the uncertainty on the determined values should be reported.
- In some of the plots, units have reciprocal part first, I personally find this quite unusual, please check the journal style.
- Readers, not very familiar with the local mass transport resistance may benefit from a brief description of how/why R_{total} and R_{other} can be determined from pressure dependent current voltage curves in S3.
- In plot 4a the interfacial and bulk oxygen permeation resistance of the ionomers are presented but could perhaps benefit from a log scale or inset to more clearly show the bulk data. A slightly clearer explanation of how the data is treated to obtain these values from the bulk and interfacial resistances in the SI would be beneficial. Data such as that in Sup Fig 6b shows the relative influence of the interface and bulk resistances on cell performance more intuitively and the authors may wish to consider including it in Fig 4 for clarity if possible.
- For LSVs on Pt(111) what are the estimated uncertainty on the experimental and MD surface coverages of R-SO₃? What is the cause of the altered redox behaviour at higher potentials in the Nafion LSV (Pt oxidation peaks) compared to the other case, is this not potentially more indicative of the influence of Nafion on ORR performance?

Finally, thank you to the authors and editor for the opportunity to review a very interesting manuscript.

Best regards

Dr Graham Smith

Senior Research Scientist

National Physical Laboratory

Reviewer #2 (Remarks to the Author):

This manuscript characterizes the property and performance of highly oxygen-permeable ionomer in PEFC both experimentally and computationally. It shows that the high solubility and permeability of the new type of ionomer incorporating a ring-structured backbone matrix is the underlying reason. While the work is worth being shared with the community, the inadequacy in originality make it unsuitable to be published in Nature Communications. The effect of ring-structured backbone on the performance of the PEFC was reported before (as early as 2013, for example, Yamada K, Hommura S, Shimohira T. Effect of high oxygen permeable ionomers on MEA performance for PEFC[J]. ECS Transactions, 2013, 50(2): 1495.), and the characterization techniques, either experimentally or computationally, are also reported before by the author's group. The only novelty lies in the application of these characterization techniques to the ionomer with or w/o ring-structured backbone.

Other minor points are annotated in the manuscript for the author's reference.

Reviewer #3 (Remarks to the Author):

The paper entitled: "Role of Highly Oxygen-permeable Ionomer in Polymer Electrolyte Fuel Cells" by Jinnouchi et al. describes the synthesis and characterization of a new ionomer showing an increased oxygen permeability. The topic discussed here is relevant for a broad audience being targeted to the cost reduction and optimization of fuel cells, one of the pillars of the hydrogen economy. Physico-chemical and electrochemical characterizations are well documented and supported, nevertheless I would recommend to doublecheck the polymer (Nafion and HOPI) molecular weight data. Conclusions drawn in the paper support through different analytical methods what already reported in previous works (also, but not only, from the same Authors). I think this paper is worth to be published on Nature Comm. with some minor revisions highlighted in track-changes mode in the attached file.

Response to comments from Reviewer #1

In this manuscript the authors report a novel perfluorinated ionomer for polymer electrolyte membrane fuel cells featuring a ring-structure in the polymer backbone. The manuscript includes characterisation of the polymer's functional properties (molecular weight, equivalent weight, ...); current-voltage curves of operating cells; microelectrode experiments to determine oxygen diffusion properties; single crystal electrode studies to investigate surface interactions and molecular dynamic simulations of ionomers at the surface.

The manuscript makes the following major claims:

- That the ionomer has a higher oxygen permeability (predominantly solubility) than a traditional Nafion ionomer reducing local oxygen mass transport resistance with the main effect being at the boundaries of the ionomer thin films;
- That the ionomer exhibits less sulfonate ion poisoning of a platinum surface than a traditional Nafion ionomer increasing mass activity of the catalyst;
- That these effects are caused by the inclusion of the ring-structure in the polymer backbone which prevents layered backbone folding.

The ionomer appears novel and its increased permeability and resulting increase in cell performance when compared to Nafion, still a state-of-the-art ionomer, is impressive. As the authors highlight, other groups have developed other highly oxygen permeable ionomers (HOPI) e.g. Ref 36 [A. Rolfi] and Ref 37 [A. Katzenberg et. al.] in which the authors incorporate an alternative cyclic group into the backbone of an ionomer and make similar claims around disruption of polymer crystallinity and resulting improved oxygen permeability. Nevertheless, this work makes a substantial contribution to the development of fuel cell ionomers and contributes to the ongoing discussion around the origins of local mass transport resistance in fuel cell catalyst layers. Both of these areas are of great topical importance in the development of PEMFC. It is likely therefore, to be of great interest to both the ionomer and catalyst layer community and the wider PEMFC and electrocatalyst fields. In my opinion this work is likely, with the support other recent publications in the area, to influence the community to further develop HOPIs in order to improve the performance of PEMFC.

We thank the referee's understanding on the contributions of our study to clarification of the role of the HOPI ionomer. The changes made in the manuscript are shown as yellow in the supplementary manuscript and SI files prepared for the review purpose.

1) Generally, the manuscript supports its major claims in a convincing way. The MEA experiments reported show cells operating with a higher performance and lower local oxygen transport resistance. The microelectrode experiments performed support this, showing much lower interfacial oxygen permeation resistance for HOPI than Nafion. This is evident both in the calculated permeation resistances and the raw data. Single crystal LSVs experiments in oxygen and nitrogen show clear evidence that Pt(111) is more active when covered with HOPI than Nafion and that REDOX processes on the surface change. The hypothesis that the incorporation of the ring structure alters the folding of the polymer backbone and thereby impacts the oxygen transport at the Pt-Ionomer interface is sensible; however I don't have the required background in MD to understand if the simulation presented here has will have high enough fidelity for the claim to be fully supported without further corroborating evidence. The evidence that changes to the backbone of the polymer are responsible for changes in absorption onto the surface is slightly tenuous. Compared to Nafion both the backbone and sidechain of HOPI have changed and either could conceivably alter the sidechain/sulfonate interaction with the surface.

We have added additional discussion and experimental results to “Results and discussion” section in the main text to show that the ring-structured PDD matrix contributes to both the high oxygen permeability and high catalytic activity. Past studies on TeflonAF composed purely of the PDD (Resnick and Buck, *Modern Fluoropolymers*, edited by John Scheirs (1997)) and Aquivion with shorter sidechains (Mitsushima et al. *J. Electrochem. Soc.* **149**, A1370 (2002)) indicate that the PDD matrix has the major contribution to the high O₂ permeability. In addition, our additional measurement on the Aquivion ionomer shown in the revised Fig. 6 indicates that the higher ORR activity and lower surface coverage of sulfonate group are mainly attributed to the PDD, too. The discussion is added to the revised main text.

2) There is no chemical or direct physical characterisation of the produced polymer provided in the supporting information. It's therefore not clear how the authors directly confirmed that a co-polymer is produced. Furthermore, there is no information on the degree or type of co-polymerisation or evidence of the crystalline structure of the polymer which would support the MD simulations.

To show chemical and physical characterization of our HOPI, we have added the experimental results of gel permeation chromatography (GPC), ¹⁹F nuclear magnetic resonance (NMR), and wide angle X-ray scattering (WAXS) measurements to Fig. 2d and e and Supplementary Fig. 2. A single peak was observed in the molecular weight distribution curve for each of Nafion and HOPI as shown in Supplementary Fig. 2a. The NMR spectra of the ionomers shown in Fig. 2d and Supplementary Fig. 2b indicate the presence of both the PFSA and PDD matrices in the HOPI. By these two facts, we judge that the PFSA and PDD were co-polymerized.

3) In the HOPI the sulfonic acid side chain is different from that in Nafion. Such short side chain ionomers / membranes are well known e.g. 3M and Aquivion ionomers and the properties are known to differ substantially from Nafion. It's therefore somewhat unclear to what extent improvements (especially relating to interaction with the Pt-111 surface) are caused by changes to the sidechain vs the backbone. Comparison experiments (or at least a literature comparison of properties such as oxygen solubility) between the performance of HOPI and an equivalent short side chain ionomer would be valuable.

Responses to this comment were given in 1).

4) Generally, the discussion would benefit from links to relevant literature so that measured values, for instance permeability, are presented in the context of other work (e.g. other reports of measured properties on Nafion, Aquivion and other HOPI) to highlight the relative significance of the improvement reported.

Comparisons with the Nafion and Aquivion ionomers were added as written in the responses to previous comments. We have compared our results with results on other HOPIs reported by Katzenberg et al. and Rolfi et al. The discussion was added to "Results and discussion" section in the main text. The comparison indicates that our HOPI exhibits more noticeable reduction of the gas transport resistance in the MEA than Katzenberg's

HOPI. The comparison of the gas permeability indicates that the enhancement of the permeability by our HOPI is similar to the result reported by Katzenberg et al. and is larger than the result reported by Rolfi et al. The comparisons indicate that our HOPI exhibits similar or superior performance to past HOPIs. We, however, stress that main purpose of this study is to show that both local oxygen transport and catalytic activity are simultaneously enhanced by the HOPI and to address the mechanism of the improved performances.

5) Many of the experimental properties of the ionomer reported, such as the mass transport resistance, solubility and diffusion coefficient, are reported without error bars or other indication of uncertainty. There is no indication that repeats were performed nor is reproducibility of the microelectrode experimental technique discussed.

To show the reproducibility and error bars, we have carried out two new sets of experiments. Because we needed to execute the quasi-steady state measurements by using a different fuel cell testing system, we carefully prepared new MEA samples and carried out all required electrochemical measurements on the new samples. We have also carried out additional thick film microelectrode experiments and molecular dynamics simulations to show the error bars. For thin film microelectrode experiments, because we already carried out experiments on many samples, we have calculated the error bars of the gas transport resistances from the fitting errors assuming normal error distributions. All new results and error bars are shown in the revised figures. The new experiments indicate that the ionic resistivity of the catalyst layers with the HOPI is higher than that with the Nafion ionomer. The result disagrees with our previous result. However, the difference makes only minor changes in the description on the ionic resistivity. The newly obtained gas diffusion resistance and their error bars indicate that the low gas permeation resistance realized by the HOPI is reproducible and reliable.

6) The electrochemical characterization performed on MEA is carried out potentiodynamically with rather high sweep rates (20 mV s⁻¹). The vast majority of MEA testing is carried out under steady state/pseudo-steady state conditions as this allows time for the cell to equilibrate. Ideally MEA testing would be repeated under conditions more normal for cell testing.

We have carried out all experiments on MEAs under the pseudo-steady state condition.

In new measurements, the potential was varied from 0.3 to 0.9 V at 3 min intervals of 0.1 V. Current was averaged over the 3 min potential hold. This pseudo-steady state data was shown as the current-voltage curves in the revised Fig. 3a. We stress that that the pseudo-steady state current-voltage curves do not exhibit significant differences from the potentiodynamic ones as illustrated in the new Supplementary Fig. 4. Because the potentiodynamic measurement eases the determinations of the current density at arbitrary IR-corrected voltage and limiting current density, we adopted the potentiodynamic measurement to determine the mass activity and gas transport resistance R_{other} . The information is written in Section S3 in the revised Supplementary Information.

7) There is some key experimental information missing, though presumably available in the author's earlier works, for instance the geometry of the microelectrode and information on electrode conditioning and electrolyte used in measurements. It's recommended the authors review the experimental sections reported add some key information to the SI if space permits.

We have added experimental information on the geometry of the microelectrode, electrode conditioning, the cell and electrolyte used in the measurements to Section S4 in the SI. Further details are written in our previous publication, ref.21.

8) The quality of text in Fig 4 is low in my copy, please check it's uploaded correctly.

We have checked that the quality of the text in the revised Fig. 4 is high in the PDF file.

9) When measuring water uptake (λ) please provide more experimental details and information on data processing (any averaging of up and down sweeps etc.).

The experimental details and data processing were added to Section S2 in the revised Supplementary Information. Supplementary Fig. 3 was also revised to clearly show each data point.

10) For MEA testing please report all relevant conditions, e.g. those identified in the JRC's Fuel Cell Harmonised Testing Protocol.

We have added relevant information on the cell temperature, gaseous temperatures at inlet of the single cell, pressures, and the potential in both cell conditioning and measurements following the information written in the EU harmonized testing protocols and protocols recommended by fuel cell commercialization conference of Japan (FCCJ) in Section S3 in Supplementary Information. We have also added two references of the EU and FCCJ protocols in the main text (“EU harmonized test protocols for PEMFC MEA testing in single cell configuration for automotive applications. 2015, <https://ec.europa.eu/jrc/en/news/new-harmonised-test-protocols-pem-fuel-cells-hydrogen-vehicles>” and “Web site of the Fuel Cell Commercialization Conference of Japan (FCCJ) (in Japanese). 2007, http://fccj.jp/pdf/23_01_kt.pdf”).

11) For measurements of local oxygen transport resistance: how is the partial pressure of the oxygen adjusted? Partial pressures should ideally be reported as pressures in MPa or kPa not %.

We should have described as “volumetric fraction” not as “partial pressure”. The volumetric fractions of H₂ and O₂ in the dry anode and cathode gases, respectively, were controlled to 20% and 1%, respectively, by mixing N₂ gas. The dry gas flow rates were set to 500 and 2000 standard cubic centimeter per minute for the anode and cathode, respectively. The total pressure was controlled by back-pressure regulators. We have described this information to the Supplementary Information.

12) In Sup Fig 5 Nyquist plots should be on isometric axes.

The Nyquist plots obtained on the new MEA samples are shown on isometric axes in the revised Supplementary Fig. 7.

13) In Sup Fig 7 more information about the quality of the fitting and the uncertainty on the determined values should be reported.

As shown in the revised Supplementary Fig. 9, the analytical function accurately reproduces the measured current density within the root mean square errors of 0.19 and 0.27 mA cm⁻² for the Nafion ionomer and HOPI, respectively. The error propagation

analysis indicates that the uncertainty caused by the small fitting error is less than 10% both for the solubility and diffusion coefficients. The results of the error propagation analysis are shown in the revised Supplementary Fig. 9. We also have carried out additional experiments on new samples and confirmed the reproducibility of the results. The differences in two experiments are shown as error bars in the revised Fig. 4 **c** and **d**. We have added the information in the revised Supplementary Information.

14) In some of the plots, units have reciprocal part first, I personally find this quite unusual, please check the journal style.

We have corrected the problematic style.

15) Readers, not very familiar with the local mass transport resistance may benefit from a brief description of how/why R_{total} and R_{other} can be determined from pressure dependent current voltage curves in S3.

The pressure-dependent R_{mol} in R_{total} stems from intermolecular diffusion in relatively large pores, where diffusion coefficient is inversely proportional to the total pressure. There is also a pressure-independent resistance R_{other} , which is attributed to Knudsen diffusions in small pores and diffusions in condensed phases, such as ionomer and liquid water. We have added the explanation in “Method” section in the revised main text.

16) In plot 4a the interfacial and bulk oxygen permeation resistance of the ionomers are presented but could perhaps benefit from a log scale or inset to more clearly show the bulk data.

Done.

17) A slightly clearer explanation of how the data is treated to obtain these values from the bulk and interfacial resistances in the SI would be beneficial. Data such as that in Sup Fig 6b shows the relative influence of the interface and bulk resistances on cell performance more intuitively and the authors may wish to consider including it in Fig 4 for clarity if possible.

We have moved the Supp Fig. 6b in the unrevised SI to the Fig. 4**a** in the revised main text. We have modified the explanation for the data treatment to determine the bulk and

interfacial resistances in the revised SI.

18) For LSVs on Pt(111) what are the estimated uncertainty on the experimental and MD surface coverages of R-SO₃⁻? What is the cause of the altered redox behavior at higher potentials in the Nafion LSV (Pt oxidation peaks) compared to the other case, is this not potentially more indicative of the influence of Nafion on ORR performance?

To show the uncertainty of the surface coverage of R-SO₃⁻, we have carried out additional experiments and MD simulations. The new results are shown in the revised Fig. 5f and 6d in the main text. As discussed in the literatures (for example, Markovic and Ross, *Surf. Sci. Rep.* **117** (2002); Subbraman et al., *ChemPhysChem*, **10**, 2825 (2010); Kodama et al., *J. Electrochem. Soc.*, **161**, F649 (2014)), the altered redox behavior relevant to the Pt oxidation is caused by the presence of the adsorbed R-SO₃⁻. On Pt sites covered by the anion species, oxide formation is suppressed by the site blocking effect. Similar behavior is observed in the H₂SO₄ solution, where (H⁺)SO₄²⁻ is specifically adsorbed on the surface (Jinnouchi et al., *PCCP*, **14**, 3208 (2012)). Hence, in these electrolytes, the amount of the oxide can become an indirect indicator of the ORR as the referee expects. However, as reported in the past studies, the surface coverage of the site blocking anion more directly describes the site blocking effect. Therefore, we use the surface coverage of R-SO₃⁻ in this study.

19) Finally, thank you to the authors and editor for the opportunity to review a very interesting manuscript.

We also thank the reviewer for many valuable comments. We believe that the revised manuscript more convincingly presents our conclusion.

Response to comments from Reviewer #2

1) This manuscript characterizes the property and performance of highly oxygen-permeable ionomer in PEFC both experimentally and computationally. It shows that the high solubility and permeability of the new type of ionomer incorporating a ring-structured backbone matrix is the underlying reason. While the work is worth being shared with the community, the inadequacy in originality make it unsuitable to be published in Nature Communications. The effect of ring-structured backbone on the performance of the PEFC was reported before (as early as 2013, for example, Yamada K, Hommura S, Shimohira T. Effect of high oxygen permeable ionomers on MEA performance for PEFC[J]. ECS Transactions, 2013, 50(2): 1495.), and the characterization techniques, either experimentally or computationally, are also reported before by the author's group. The only novelty lies in the application of these characterization techniques to the ionomer with or w/o ring-structured backbone.

We appreciate for many valuable comments. The changes made in the manuscript are shown as yellow in the supplementary manuscript and SI files prepared for the review purpose.

Although highly oxygen permeable ionomers (HOPIs) were reported in past studies, their practical applications in fuel cells started only recently. Mechanisms underlying high performances of the HOPIs are not clear. In addition, as discussed in our manuscript, dominant factor of the high local oxygen transport resistance of the Nafion ionomer remains controversial.

In this study, we have newly clarified two major roles of the highly oxygen permeable ionomer: reduction of the interfacial gas transport resistance and enhancement of the ORR activity. Both improvements are attributed to changes in the interfacial structure between the Pt electrode and ionomer. Our new discovery indicates that high ORR activity and high O₂ permeation are compatible properties. In addition, detailed MD analyses on the Nafion ionomer and HOPI indicate that oxygen permeation through the ionomer/Pt interface remains the rate limiting process in the HOPI. The information can provide new guiding principles to design further high-performance catalyst layer structures, for example, by modifying molecular structures of ionomer and/or by adopting mesoporous support material to mitigate direct ionomer-catalyst contact.

Correct understanding of the microscopic mechanism is always significant for designing new materials. The significance can be exemplified by Pt alloy catalysts. Active Pt alloy

catalysts were known in this community before 2000. However, intensive experimental and theoretical studies on atomistic and electronic structures have revealed the mechanisms, providing guiding principles to design new shape controlled alloy catalysts, later. Many of these fundamental studies have been reported in high impact journals published by Nature publishing group.

We believe that Nature Communications is the appropriate platform to communicate the present observations to the relevant community.

Other minor points are annotated in the manuscript for the author's reference.

Response to each comment is listed below.

2) Efficiency is usually determined by working voltage for fuel cell. Activity could be a better word.

We have changed “efficiency” to “catalytic activity”.

3) Here performance and durability are mentioned parallelly.

We have change “performance” to “power density. We have carefully revised the wording in other parts.

4) Here durability in included in performance.

By the revision in 3), we think that the problem in the wording was solved.

5) This comment/conclusion is objective. However, in the results and discussion part, this work takes the view of “interface dominated mechanism” without further discussion.

We take the view of “interface dominated mechanism” because our experimental and theoretical results presented in this study indicate that the interfacial resistance is dominant. We have added more explicit descriptions of the conclusion in “Results and discussion” and “Conclusion” sections in the revised manuscript.

6) This is a weak gap.

7) The idea of “Ring-structured monomer” was presented in previous paper. What is the key contribution of this work?

As in the responses to 1), a key contribution of our study is to clearly demonstrate the concept that both interfacial permeability and catalytic activity can be simultaneously enhanced by the HOPI. Both enhancements are attributed to modifications in the ionomer/Pt interfacial structure. As discussed in “Introduction” section, in electrocatalytic materials, one property is often in a trade-off relationship to the other in fuel cells. Therefore, our discovery is highly significant in this community. Another significant conclusion, which was not clearly written in our previous manuscript, is that the interfacial oxygen transport resistance remains a rate limiting process of the oxygen permeation through the HOPI. The knowledge provides significant guiding principles to realize high-performance cathode catalyst layer, for example, by optimally locating catalysts, ionomers and support materials in the catalyst layer and/or by designing molecular structure of ionomer. To stress the point, we have revised “Introduction” section and added new Fig. 5 e, which shows that the HOPI/Pt interface still needs improvement.

8) The interfaces include gas/ionomer interface and ionomer/Pt interface. Which one is significantly improved?

Our MD simulation indicates that O₂ permeation at ionomer/Pt interface is significantly improved by HOPI. Since this point is not clearly mentioned, we have revised “Results and discussion” and “Conclusion” sections.

9) R_{other} is measured by limiting current method. What is the quantitatively relations between R_{other} and the voltage loss. For the case of 90% RH, R_{other} of Nafion and HOPI differs, while the performance curve coincides. Why?

We considered that the contradicting result is because of errors in the MEA measurements. We have carried out additional experiments to obtain more reliable current-voltage curves and R_{other} as well as error bars. The new results shown in the revised Fig. 3 indicates that the limiting current density of the HOPI at 90%RH is higher than that of the Nafion ionomer. The result is consistent with the measured R_{other} .

10) How to understand the non-monotonic trend of resistance with RH?

We consider that the non-monotonic trend in the result is because of the uncertainty in the MD simulation. Oxygen permeation through the ionomer is a rare event, which is difficult to be quantitatively examined by the MD simulations. Even under the increased pure oxygen pressure, a few oxygen molecules can permeate the Nafion ionomer within the 150 ns MD simulations. To obtain more reliable results, we have carried out two additional MD simulations. The new result shows more monotonic trend, and the averaged resistance of the HOPI gradually decreases with the decrease in the RH. The results are shown in the revised Fig. 5.

11) Please show the error bars of the data.

To show the error bars, we have carried out additional experiments and MD simulations. Error bars have been added in the revised Figs. 4, 5 and 6.

12) The O₂ solubility and diffusion coefficient can also be calculated by MD simulations. Please show the simulation value and compare it with the experimental value.

We have carried out MD simulations to calculate diffusion coefficients and solubilities of oxygen in Nafion and HOPI. The results reasonably agree with the experimental results. The results are shown in the revised Fig. 5. The simulation methods were also added to Methods and Section S5 in SI.

13) “thin films” in figure caption, while “thick film” in figure b, c. This is not consistent.

The figure caption was wrong. We have changed “thin films” in the figure caption to “thick film.”

14) The dimension/unit of interfacial oxygen permeation resistance in “figure a” and “figure d” is not consistent. How to compare the results?

For better comparison, we have changed the unit in the simulated result to s m^{-1} similarly to the unit in the experimental result.

Response to comments from Reviewer #3

1) The paper entitled: "Role of Highly Oxygen-permeable Ionomer in Polymer Electrolyte Fuel Cells" by Jinnouchi et al. describes the synthesis and characterization of a new ionomer showing an increased oxygen permeability. The topic discussed here is relevant for a broad audience being targeted to the cost reduction and optimization of fuel cells, one of the pillars of the hydrogen economy. Physico-chemical and electrochemical characterizations are well documented and supported, nevertheless I would recommend to doublecheck the polymer (Nafion and HOPI) molecular weight data. Conclusions drawn in the paper support through different analytical methods what already reported in previous works (also, but not only, from the same Authors). I think this paper is worth to be published on Nature Comm. with some minor revisions highlighted in track-changes mode in the attached file.

We thank referee for the understanding on the contributions of our study on the HOPI. The changes made in the manuscript are shown as yellow in the supplementary manuscript and SI files prepared for the review purpose.

2) I would recommend to compare this figure with a literature reference (please cite it in the reference list). It is commonly recognized that Nafion molecular weight is about (100.000 or even higher). See for example Mauritz, K.A.; Moore, R.B. Chem. Rev. 2004, 104, 4535-4585.

We have added the citation to the review paper by Mauritz and co-workers and compared the results in the revised manuscript. The molecular weight measured by the gel permeation chromatography strongly depends on the standards used for the calibration. In our study, pullulan with a range of molecular weight 1420 to 344 000 g mol⁻¹ was used, and this standard provided relatively small molecular weights comparing with the reported values. Although the absolute values are different, we can discuss the relative difference between the Nafion ionomer and HOPI. We have added the information and discussion in the main text and Supplementary Information.

3) In order to be more reader friendly I would recommend to briefly explain the root-cause of this unexpected behavior.

We have added the explanation for the mechanism of the unexpected result on the proton conduction in the main text. As discussed by Katzenberg and co-workers, PFSA domains generate pathways for proton conduction while the amorphous domains do not. Therefore, the proton conductivity of our HOPI is not higher than that of Nafion while the EW of the former is lower.

4) In Fig. 5c you reported the reference voltammogram of Pt(111) (blue dashed line). For sake of clarity, I would recommend to mention this and briefly comment.

In “Results and discussion” section in the revised manuscript, we have added an explicit description that the revised Fig. 6c contains the reference voltammogram of the Pt(111) surface without ionomer. We have also described that the redox charges attributed to the adsorption and desorption of the sulfonate anion are not observed in the reference voltammogram.

5) I guess it is a water solution. For sake of clarity it is better to make it explicit.

Yes, this is an aqueous solution. We explicitly describe it in the revised main text.

6) As above, I think it is a water solution of HCl. Better to clarify.

Done.

7) Nafion is now commercialized by Chemours.

We have changed to Chemours.

8) In order to be more reader friendly I would appreciate to make the units of measurement explicit.

9) As above. I would ask the Authors to clarify the units of measurement.

Done.

10) Please address the typo: Bipoloar plate

Done.

11) Being cited in the text and for sake of comparison, I think would be helpful to report the structure and the available data of PFMMD-based ionomer (ref. 37 of this paper)

The molecular structure of the PFMMD-based ionomer and its available data are added to Fig. 1.

12) As mentioned in the text please doublecheck the Nafion Mn in literature

Done.

REVIEWER COMMENTS

Reviewer #1 (Remarks to the Author):

Thank you for the updates to the manuscript. The changes address my major concerns from the first review about the influence of the headgroup, the characterisation of the synthesised HOPI, uncertainty on measurements, and cell testing.

Best regards
Graham Smith
National Physical Laboratory

Reviewer #2 (Remarks to the Author):

Thanks for your efforts in revising the manuscript substantially, including checking the repeatability, adding new test results, and expanding the discussion. The quality of the manuscript has been enhanced greatly. I believe now that the work is worthy being shared with the ionomer and catalyst layer R&D community.

Reviewer #3 (Remarks to the Author):

The comments and recommendations raised in the review run have been addressed by the Authors; the open questions were properly answered. I think the paper is now suitable for publication on Nature Comm.